# Graph-enhanced deep learning for diabetic retinopathy diagnosis: A quality-aware and uncertainty-driven approach

Zarin Akter, Jawad Ibn Ahad, Md. Mutasim Farhan, Riasat Khan*

Electrical and Computer Engineering, North South University, Dhaka, Bangladesh

* riasat.khan@northsouth.edu

## Abstract

Diabetic retinopathy (DR) is a leading cause of vision impairment, which significantly impacts working-class populations, necessitating accurate and early diagnosis for effective treatment. Traditional DR classification relies on Convolutional Neural Network (CNN)-based models and extensive preprocessing. In this work, we propose a novel approach leveraging pre-trained models for feature extraction, followed by Graph Convolutional Networks (GCNs) for refined embedding representation. The extracted feature vectors are structured as a graph, where GCN enhances embeddings before classification. The proposed model incorporates quality assessment by predicting a confidence score through a dedicated fully connected layer, trained to align with ground truth quality using binary cross-entropy loss. Uncertainty estimation is achieved by calculating the variance across multiple stochastic passes, providing a measure of the model's prediction reliability. We evaluate the proposed DR detection approach on APTOS2019, Messidor-2, and EyePACS datasets, achieving superior performance over state-of-the-art methods. Using MobileViT as the main feature extractor, we reached a remarkable 98.45% accuracy, 98.45% F1-Score, and 98.06% Kappa on the APTOS2019 dataset. The DenseNet-169 proved to be the best backbone of the pipeline for the Messidor-2 dataset, with an accuracy of 94.90%, F1-Score of 94.87%, and Kappa of 93.63%. Additionally, for external validation, the model demonstrated strong generalization capability on the EyePACS dataset, where DenseNet-169 achieved 97.38% accuracy, 97.37% F1-Score, and 96.72% Kappa, while MobileViT obtained 96.02% accuracy, 96.02% F1-Score, and 95.03% Kappa. Our innovative architecture incorporates uncertainty estimation and quality assessment techniques, enabling accurate confidence scores and enhancing the model's reliability in clinical environments. Furthermore, to strengthen interpretability and facilitate clinical validation, Grad-CAM heatmaps were employed to demonstrate the significance of different input regions on the model's predictions.

**Data availability statement:** The datasets analyzed in this study are publicly available. The APTOS 2019 dataset can be accessed from Kaggle at https://www.kaggle.com/competitions/aptos2019-blindness-detection. The Messidor-2 dataset is available from ADCIS at https://www.adcis.net/en/third-party/messidor2. The implementation codes can be found at: https://github.com/mfar201/diabetic_retinopathy_classification_gcn.

**Funding:** The author(s) received no specific funding for this work.

**Competing interests:** The authors have declared that no competing interests exist.

## Author summary

Diabetic retinopathy is a diabetes-induced severe eye condition that leads to permanent blindness if not treated at an early stage. In this study, we introduce a new method that uses pre-trained models to extract features, which are then refined by Graph Convolutional Networks (GCNs) for better embedding representation. These feature vectors are structured as a graph, where the GCN improves the embeddings before classification. Our model assesses image quality by predicting a confidence score and estimates prediction reliability by calculating variance from multiple stochastic passes. We tested our DR detection approach on three datasets: APTOS2019, Messidor-2, and EyePACS, outperforming current state-of-the-art methods. We evaluate the proposed DR detection approach on APTOS2019, Messidor-2, and EyePACS datasets, achieving superior performance over state-of-the-art methods. Using MobileViT as the main feature extractor, we reached 98.45% accuracy, 98.45% F1-Score, and 98.06% Kappa on the APTOS2019 dataset. The DenseNet-169 proved to be the best backbone of the pipeline for the Messidor-2 dataset, with an accuracy of 94.90%, F1-Score of 94.87%, and Kappa of 93.63%. The model demonstrated strong generalization capability on the EyePACS dataset, where DenseNet-169 achieved 97.38% accuracy, 97.37% F1-Score, and 96.72% Kappa, while MobileViT obtained 96.02% accuracy, 96.02% F1-Score, and 95.03% Kappa.

## 1 Introduction

Diabetic retinopathy (DR) is a diabetes-induced severe eye condition that can lead to permanent blindness if not treated at an early stage [1,2]. Anyone with type 1, type 2, or gestational diabetes (suffering from diabetes while pregnant) can develop this vision-threatening disease [3]. In this condition, elevated blood sugar levels damage the tiny blood vessels in the retina, leading to swelling, leakage, and abnormal vessel formation [4]. During the early stages, DR may not be diagnosable through symptoms or very mild vision problems [4]. However, if left untreated for a long time, it can lead to partial vision loss or permanent blindness. The progression of DR can be identified in four main stages, which are: *(a)* **Mild Non-Proliferative DR (Mild NPDR):** In this stage, microaneurysms (small bulges) are seen in the retinal blood vessels. Usually, no symptoms are seen during this stage. *(b)* **Moderate Nonproliferative DR (Moderate NPDR)**: Increased microaneurysms start interfering with the regular retina blood flow. Other lesions begin to develop, such as hemorrhages and hard exudates. *(c)* **Severe Non-Proliferative DR (Severe NPDR):** The body starts to signal the formation of new and abnormal blood vessels in the retina. *(d)* **Proliferative DR (PDR)** is the severest stage of DR, where new abnormal blood vessels (neovascularization) form in the retina that are very fragile and prone to leakage. Possible vision problems during this stage are blurriness, reduced vision, and blindness [5,6].

According to the International Diabetes Federation (IDF), approximately 537 million adults (aged between 20 and 79) are living with diabetes. Their study shows

that by 2045, the number of diabetic patients will increase by 46% (approximately 763 million), which means one in eight adults will suffer from diabetes [7]. It has also been found from a study in 2021 that among the population of diabetic patients worldwide, 22.27% suffer from DR, which indicates that nearly one in four diabetic patients suffer from DR [8]. The potential rise in DR cases in the near future underscores the urgent need for reliable and effective strategies for early detection of the problem to prevent blindness.

The traditional methods for detecting DR involve a few methods, i.e., **(a) Optical Coherence Tomography (OCT):** It provides detailed cross-sectional images of the retina, thus enabling doctors to check whether it has swelled or not [9]. **(b) Funduscopy:** The eye's retina is examined with an ophthalmoscope to check for different types of lesions, such as microaneurysms, hemorrhages, etc. [10]. However, the traditional tools needed for the detection of DR are costly and time-consuming, which can be a barrier for many healthcare providers. The recent advancements in the field of artificial intelligence (AI), particularly in the deep learning field, have made it possible for researchers to address these challenges [11]. **(c) Deep learning predictive modelings:** Convolutional Neural Networks (CNNs), Vision Transformers (ViTs), and Graph Neural Networks (GNNs) have shown significant promise in medical diagnosis [12–14] These techniques can be used to classify DR accurately and reliably. To date, AI methods excel in binary DR classification but struggle with reliable multiclass DR predictions.

To address this gap, we propose a novel pipeline, summarized in Fig 1, that integrates a graph convolutional network (GCN) with pre-trained models as feature extractors (FE) for DR prediction using real fundus images. Existing research often relies on extensive data preprocessing, making the pipeline computationally heavy and sensitive to real-world images. These preprocessing steps can degrade image authenticity, which is particularly critical for fundus images. To overcome this issue, a novel approach has been proposed in this work that directly utilizes fundus images with only basic resizing and normalization. It is important to distinguish this minimal preprocessing from the common practice of online

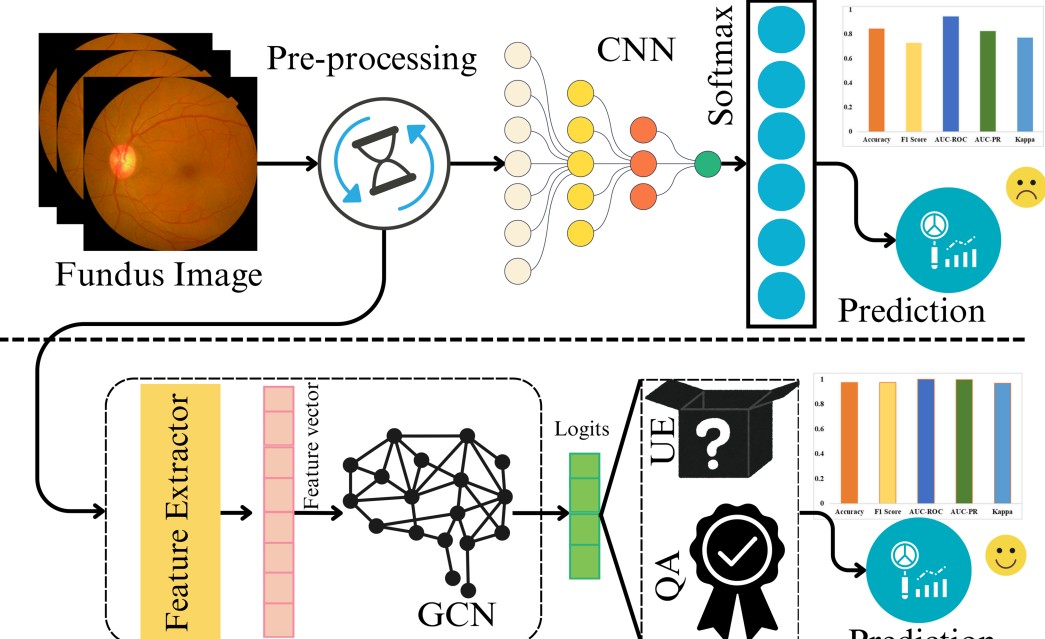

**Fig 1. Traditional DR classification (up) relies on CNN-based models with extensive preprocessing on raw data [15,16].** In contrast, the proposed novel strategy (down) utilizes pre-trained models for feature extraction (FE). The generated feature vectors (FV) are refined using a Graph Convolutional Network (GCN) and subsequently leveraged for classification, quality assessment (QA), and uncertainty estimation (UE).

data augmentation (e.g., random rotations and flips), which we employ during training to improve model generalization. To validate the robustness of our primary, preprocessing-free pipeline, we conduct a comparative analysis against versions of our model that incorporate intensive preprocessing steps, such as CLAHE and Ben Graham's method. Our results confirm that the applied additional steps are unnecessary to achieve state-of-the-art performance with our proposed framework. Additionally, we explore an innovative training strategy that incorporates quality assessment (QA), uncertainty estimation (UE), and classification losses, enhancing model reliability.

This study aims to achieve robust and accurate multiclass DR classification and demonstrates the following contributions:

- Introduces a novel framework that integrates GCN with FE for multiclass DR classification using fundus images without excessive preprocessing.
- Employs a unique loss formulation that combines QA and UE losses with classification loss, enhancing model reliability and performance. QA evaluates the reliability of predictions using a fully connected layer trained with binary cross-entropy loss, and UE quantifies prediction confidence by calculating the variance across multiple stochastic passes through the network.
- Proposes a strategy that preserves image authenticity, enhancing robustness to real-world variations.
- Utilizes Grad-CAM heatmaps to strengthen interpretability and facilitate clinical validation and the significance of different input regions on the model's predictions.

The rest of the article is structured into distinct sections as follows: Sect 2 reviews related works; Sect 3 describes the methodology of the proposed graph convolutional networks-based diabetic retinopathy detection system; Sect 4 shows the experimental setup, evaluation metrics; Sect 5 presents the test results, compares the performance of our proposed model against established benchmarks, and includes an ablation study; Sect 6 provides a discussion; and Sect 7 concludes the study.

## 2 Related works

The traditional techniques needed to detect DR are inefficient and time-consuming, which can be a barrier for many healthcare providers. Early DR identification often varies depending on the ophthalmologist's subjective interpretation. The availability of experienced doctors and expensive tools affects the detection process. In recent years, computer vision-based frameworks have been employed for DR classification.

### 2.1 Diabetic retinopathy prediction

Mondal et al. [15] proposed an ensemble model using DenseNet101 and ResNeXt for DR classification, achieving 96.98% accuracy, but faced challenges with class imbalances in multiclass scenarios. Tokuda et al. [16] used a U-Net with EfficientNet6 for DR diagnosis, focusing on retinal hemorrhages, achieving sensitivity (0.812–1.0) and specificity (0.888–1.0), but the model's generalizability to diverse datasets and real-time deployment was not thoroughly addressed. Mohanty et al. [17] used a hybrid VGG16-XGBoost and DenseNet121 for DR detection in the APTOS2019 dataset, with DenseNet121 achieving 97.30% accuracy, outperforming the hybrid model (79.50%), but the study did not explore the model's scalability or performance on more diverse datasets. Arora et al. [19] introduced an innovative deep learning framework leveraging EfficientNetB0 and CNN layering for accurate diabetic retinopathy diagnosis. Their model, trained on 35,108 retinal images, achieved an impressive 86.53% average accuracy and a 0.5663 loss rate. This robust computational approach offers precise and dependable classification of DR severity levels. Yadav et al. [20] developed a framework combining Modified Inertia Weight Particle Swarm Optimization (MIWPSO) and Fuzzy C-Means (FCM) for diabetic retinopathy image segmentation. This method achieved a remarkable 98.42% accuracy, significantly enhancing diagnostic capabilities by

effectively eliminating noise and precisely segmenting medical images. Herrero-Tudela et al. [21] applied ResNet-50 on APTOS-2019, EyePACS, and DDR datasets, achieving 94.64% accuracy, 0.94 QWK on APTOS-2019, and lower metrics on others. Explainable AI (SHAP) improved interpretability, but dataset imbalance and lack of multimodal integration remained challenges. These limitations motivated us to address imbalanced multiclass DR classification. Akhtar et al. [18] proposed RSG-Net, a CNN for diabetic retinopathy grading on the Messidor dataset, achieving 99.36% accuracy, an F1-score of 0.994, a specificity of 99.79%, a sensitivity of 99.41%, an AUROC of 0.9998, and an AUPR of 0.994. The model surpasses state-of-the-art methods, with future directions including multi-dataset validation, stronger regularization, ensemble integration, and refined augmentation to enhance generalizability.

## 2.2 Feature extractor backbone

Inamullah et al. [22] proposed an ensemble CNN with augmentation techniques for DR, achieving 91.06% accuracy, 95.01% sensitivity, and 98.38% specificity, but the study did not address the model's performance on real-world, diverse clinical datasets or its interpretability. Macsik et al. [23] fused Xception and EfficientNetB4 models for DR classification, using CLAHE and augmentation, achieving 96.4% accuracy on DDR and 94.5% accuracy on APTOS2019. The authors did not address the model's robustness across different populations or its real-time applicability. Elsharkawy et al. [24] introduced Fused-AETNet, a VAE-Transformer framework integrating OCT biomarkers for DR detection. On 481 subjects, it achieved 93.08% accuracy, 93.33% precision, 96.00% recall, 94.48% F1-score, 96.70% AUROC, and a high Kappa. Future work includes multi-stage DR grading, 3D OCT biomarkers, uncertainty quantification, and clinical deployment. Rieck et al. [25] proposed a Transformer–CNN hybrid (EfficientNet-B4 + Swin Transformer V2) on the EyeDisease dataset, achieving 76.40% accuracy, 81.91% balanced accuracy, F1-score 76.65%, AUROC 0.96, AUPR 0.78, and Kappa 0.71. The model showed strong generalization, with future work targeting external validation, multimodal integration, and improved interpretability. Shaban et al. [26] employed a deep CNN with 18 convolutional units and multiple fully connected layers (FCL) for fundus image analysis, achieving 89% accuracy and a 0.915 kappa score, but faced generalization challenges due to data augmentation and class imbalance handling. Advanced deep learning models have shown positive direction for DR prediction, but they fail to grasp complex patterns, even though using extensive image preprocessing.

## 2.3 Graph neural network

Hai et al. [2] introduced the DRGCNN model for DR grading, leveraging GNNs and balanced EyePACS and Messidor-2 datasets, achieving kappa values of 86.62% and 86.16%. However, the study did not assess the model's performance on larger, more diverse datasets or real-time deployment scenarios. Feng et al. [6] proposed a hybrid CNN-GNN model for DR grading, achieving 95.6% and 94.3% accuracy on APTOS2019 and Messidor-2 datasets, respectively. Challenges include dataset diversity, computational demands, and comorbidity effects. The study did not explore the model's generalization to the real world. Sundar and Sumathy [27] proposed a hybrid Graph Convolutional Network (HGCN) for DR classification, achieving 90.34% accuracy on EyePACS and a 6.59% accuracy improvement over DenseNet. Challenges include clinical validation and dataset imbalance. Zhang et al. [28] proposed a Deep Graph Correlation Network (DGCN) for DR grading without manual annotations, integrating convolutional and graph neural networks. The model achieved 89.9% accuracy, 88.2% sensitivity, and 91.3% specificity on EyePACS-1 dataset, and 91.8%, 90.2%, and 93.0% on Messidor-2. Challenges remain in sensitivity performance and real-world clinical deployment. Cheng et al. [29] developed a multi-label classification model based on Graph Convolutional Networks (GCN) for analyzing fundus images. The model achieved an F1-score of 0.808 and an AUC of up to 0.986. Despite its impressive performance, the model faces challenges like dataset imbalance and detecting small lesions.

After reviewing the recent articles on automatic DR, it can be concluded that most of the works focused on extensive data preprocessing techniques, multiclassification performance remains a challenge in many studies, the majority of the

articles only use a single fundus image dataset, and very few of them employ explainable AI techniques. The challenges of relying on extensive preprocessing and the suboptimal performance of complex models in DR prediction motivated us to find a feasible solution. In this study, we address these gaps by introducing an unprecedented strategy that minimizes preprocessing while achieving a significant performance improvement. Table 1 summarizes the contributions and research gaps of recent DR studies applying CNN/backbone and GNN-based methods.

## 3 Methodology

In this section, we present the problem formulation, outline our solution strategy, detail the model selection process, and describe the pipeline construction, including dataset utilization and the training framework.

### 3.1 Ethics statement

This study adhered to ethical guidelines for medical AI research, using publicly available datasets (APTOS2019, Messidor-2, and EyePACS) that comply with data privacy regulations. No personally identifiable information was used, and all experiments were designed to improve healthcare accessibility while reducing bias. The goal is to support, not replace, clinicians in diagnosing DR. Future work will address additional ethical concerns, including fairness, transparency, and accountability, to ensure the model's responsible development and deployment in clinical settings.

**Problem Formulation:** Let, the input dataset $\mathcal{D}$ consists of retinal images $x_i \in \mathbb{R}^{H \times W}$, where $H$, and $W$ represent the height and width, respectively. We aim to predict DR from input images $x$ and classify into 5 classes using a backbone $f$ that extracts feature vectors (FV) $\mathbf{z} = f(x) \in \mathbb{R}^d$, where $d$ denotes the feature dimension. A graph $G = (\mathcal{V}, \mathcal{E})$ is constructed, where nodes represent the features $\mathbf{z}$ and edges are based on spatial and semantic distances. A GCN refines the feature embeddings through a series of graph-based layers. The refined embeddings $\mathbf{h}$ are then passed through a softmax classifier $\hat{\mathbf{y}} = \text{softmax}(\mathbf{W}_{cls}\mathbf{h} + \mathbf{b}_{cls})$, where $\hat{\mathbf{y}} \in \mathbb{R}^5$ corresponds to the predicted class probabilities. Additionally, the uncertainty of predictions is modeled by performing $T$ stochastic passes through the network, and the final prediction is obtained as the mean of the outputs, with uncertainty quantified as the variance across the passes. The objective is to create a robust pipeline with a pre-trained model $f$ and GCN for accurate multiclass DR prediction. The pipeline is shown in Fig 2, where two layers of GCN refine the embeddings, resulting in the final embedding $h^{(2)} \in \mathbb{R}^{256}$. This final embedding is then passed through two fully connected layers (FCLs), producing predictions $\hat{y} \in \mathbb{R}^{256 \times 5}$ for classification and $\hat{q} \in [0, 1]$ for

**Table 1**. **Related work on diabetic retinopathy (DR): Contributions, datasets, and gaps.**

| Ref. | Contribution | Dataset(s) | Limitation / Gap |
|---|---|---|---|
| [2] | DRGCNN (GNN-based DR grading) with balanced sets | EyePACS, Messidor-2 | Not assessed on larger/diverse datasets; no real-time study |
| [6] | Hybrid CNN–GNN for DR grading | APTOS-2019, Messidor-2 | Dataset diversity, computational demand, comorbidity effects; real-world generalization not explored |
| [15] | Ensemble (DenseNet101 + ResNeXt) for DR classification | DIARETDB1 | Class-imbalance issues in multiclass settings |
| [16] | U-Net with EfficientNet6 targeting retinal hemorrhages | Collected fundus images | Generalizability and real-time deployment not addressed |
| [17] | DenseNet121 vs. hybrid VGG16–XGBoost on APTOS2019 | APTOS-2019 | Scalability and performance on diverse datasets not explored |
| [21] | ResNet-50 with SHAP explainability | APTOS-2019, EyePACS, DDR | Dataset imbalance; lack of multimodal integration |
| [23] | Xception + EfficientNetB4 with CLAHE + augmentation | DDR, APTOS-2019 | Robustness across populations and real-time applicability not addressed |
| [28] | Deep Graph Correlation Network (DGCN), no manual annotations | EyePACS-1, Messidor-2 | Sensitivity and clinical deployment remain challenging |

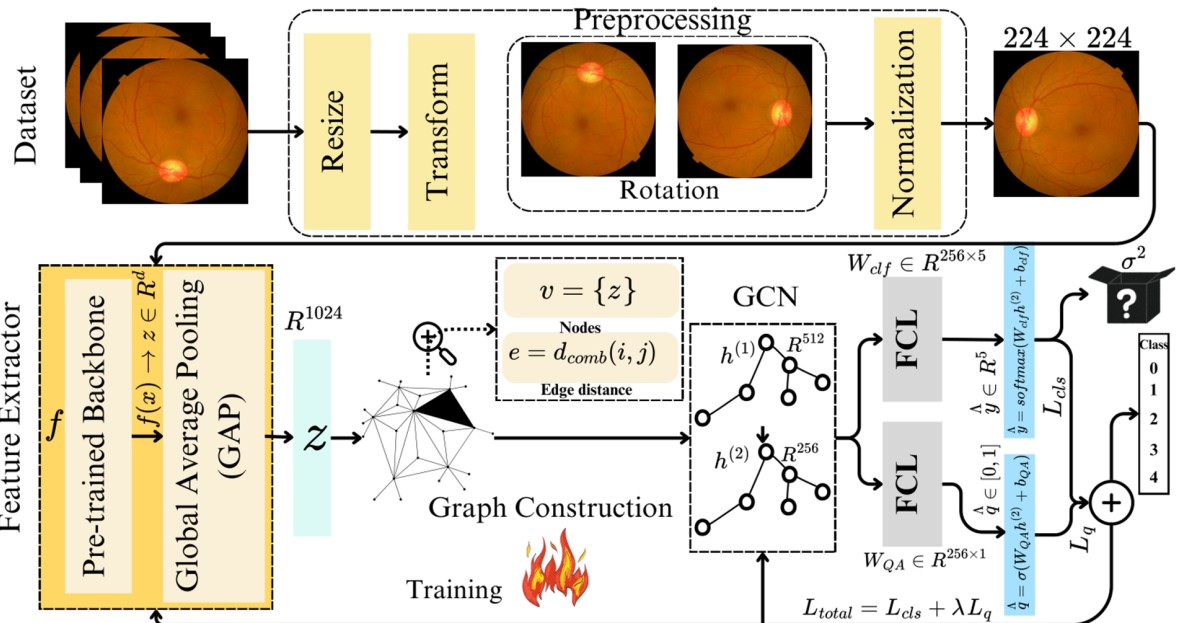

**Fig 2**. *Model architecture:* **The dataset $\mathcal{D}$ undergoes basic preprocessing (e.g., resizing, transformations, rotations) to prepare the data.** The FE function f processes each sample x ∈ $\mathcal{D}$ to generate a feature vector (FV) z ∈ $\mathbb{R}^d$. This FV is then refined to a vector in $\mathbb{R}^d$ using Global Average Pooling (GAP). A graph is constructed with nodes corresponding to FVs, and the edge distance is computed based on spatial distance $d_{sp}$ and semantic distance $d_{se}$.

quality assessment. The total loss is computed by combining the classification loss $\mathcal{L}_{cls}$ and the quality assessment loss $\mathcal{L}_q$, weighted by a hyperparameter $\lambda$. Backpropagation is used to train both the FE and GCN layers.

In this study, we incorporate two of the most novel aspects of multiclass DR classification. **(a) Quality Assessment:** Quality Assessment (QA) is used to evaluate the reliability or confidence of the model's predictions. It is typically modeled as a scalar value $\hat{q} = \sigma(W_{QA}h^L + b_{QA})$, indicating the model's prediction certainty. It is calculated using a QA head, which is a fully connected layer (FCL), with a learnable weight $W_{QA} \in \mathbb{R}^{256 \times 1}$. The goal is to minimize the discrepancy between predicted and true quality assessments during training. To quantify the quality of the predictions, a Binary Cross-Entropy (BCE) loss function is used. Given the predicted quality $\hat{q}$ and the true quality $q$, the loss function is defined as:

$$\mathcal{L}_q = \text{BCE}(\hat{q}, q) = -q\log(\hat{q}) - (1-q)\log(1-\hat{q}) \tag{1}$$

where $\hat{q}$ is the predicted quality score (between 0 and 1), and $q$ illustrates the ground truth indicating the quality of the prediction (binary: 0 or 1). This loss function encourages the model to output high-quality predictions when the true quality is high and low-quality predictions when the true quality is low. **(b) Uncertainty Estimation:** Uncertainty Estimation (UE) quantifies the model's confidence in its corresponding predictions by modeling the variance across multiple stochastic passes through the network. For a given input $x$, the model constructs $T$ stochastic passes, resulting in a set of predictions $\hat{y}^{(t)}$ for $t = 1, 2, \dots, T$. The predicted label is averaged over these passes to get the final prediction: $\bar{y} = \frac{1}{T}\sum_{t=1}^{T}\hat{y}^{(t)}$ where $\hat{y}^{(t)}$ is the prediction from the $t$-th forward pass. The uncertainty is measured as the variance across the $T$ predictions: $\sigma^2 = \frac{1}{T}\sum_{t=1}^{T}(\hat{y}^{(t)} - \bar{y})^2$. Here $\sigma^2$ represents the uncertainty of the model's prediction, with higher values indicating greater uncertainty. The uncertainty value can be used to gauge the reliability of the predictions; lower uncertainty indicates more confidence in the result. In all experiments, we set the number of Monte Carlo (MC) forward passes to $T = 10$.

Each pass keeps dropout *active* ($p = 0.3$ for the classifier head and $p = 0.2$ within the GCN layers), ensuring that different subnetworks are sampled on each pass. The final class-probability vector is computed as the arithmetic mean:

$$\bar{y} = \frac{1}{T} \sum_{t=1}^{T} \hat{y}^{(t)},$$

and the predictive uncertainty is measured as the standard deviation:

$$\sigma = \sqrt{\frac{1}{T} \sum_{t=1}^{T} \left( \hat{y}^{(t)} - \bar{y} \right)^2}.$$

To ensure reproducibility, we fixed all random seeds using `torch.manual_seed(42)`, `torch.cuda.manual_seed_all(42)`, and `np.random.seed(42)`.

### 3.2 Dataset

Three distinct fundus image-based datasets are used for this study for DR classification. **(a) APTOS (Asia Pacific Tele-Ophthalmology Society):** The APTOS2019 Blindness Detection dataset is a public collection of retinal fundus images designed to enable research on DR detection and severity classification. A single CSV file with respective labels accompanies 3,662 training images. These images were obtained in various clinical contexts and imaging parameters, showcasing differences in illumination, contrast, and clarity, thus presenting realistic diagnostic debates. **(b) Messidor-2:** Messidor-2 is a publicly available dataset that has been used extensively to develop and evaluate automated methods for DR detection and grading. It extends the original Messidor dataset, a benchmark collection of retinal fundus images of diabetic patients obtained under standardized conditions. Messidor-2 consists of 1,748 color retinal images, all with good image quality for method training and testing. Despite their uses in research, datasets are highly imbalanced. **(c) EyePACS:** EyePACS is one of the largest publicly available datasets for Diabetic Retinopathy detection. It was originally released for a Kaggle competition titled 'Diabetic Retinopathy Detection'. EyePACS provides high-resolution fundus images captured under different imaging conditions, labeled with diabetic retinopathy severity grades from 0 (no DR) to 4 (proliferative DR). The total number of images provided in the EyePACS dataset with known labels is 35,126. Statistics for all three datasets are given in Table 2.

### 3.3 Dataset preparation

We first addressed the class imbalance problem by balancing the number of samples across the five DR classes to prepare our dataset for model training. We employed an oversampling approach for minority classes, ensuring that all five

**Table 2**. **All datasets are categorized into five distinct groups based on the severity levels of DR present in the fundus images.** Here, "0" stands for No DR, "1" stands for Mild NPDR, "2" stands for Moderate NPDR, "3" stands for Severe NPDR, and "4" stands for PDR.

| Classes | DR Grades | # of Instances | | |
|---|---|---|---|---|
| | | APTOS2019 | Messidor-2 | EyePACS |
| Class - 0 | No DR | 1805 | 1017 | 25810 |
| Class - 1 | Mild NPDR | 999 | 270 | 2443 |
| Class - 2 | Moderate NPDR | 370 | 347 | 5292 |
| Class - 3 | Severe NPDR | 295 | 75 | 873 |
| Class - 4 | PDR | 193 | 35 | 708 |
| **Total** | | **3662** | **1748** | **35126** |

DR classes had an equal number of images for training. This balancing was achieved through a combination of duplicating samples using some transformations. These transformations are distinct from the intensive, dataset-wide preprocessing techniques that our primary model seeks to avoid. By following this procedure, the final class distribution across these batches was uniform, thereby mitigating bias in the model towards classes with higher initial representation.

### 3.4 Model architecture

**Backbone:** We employed fifteen backbone architectures, including four DenseNet variants: DenseNet-121 (7M), DenseNet-161 (28M), DenseNet-169 (14M), and DenseNet-201 (20M), leveraging dense connectivity for efficient feature reuse [30]. Additionally, we used three ResNet variants: ResNet50 (25M), ResNet101 (44M), and ResNet152 (60M), which introduced residual connections for improved gradient flow [31]. Lastly, we incorporated Inception V3 (23M) and Inception-ResNet-v2 (55.9M) for multi-scale feature extraction and enhanced gradient propagation [32].
For Transformer architectures, we used ViT-base (86M), Swin-base (88M), and DeiT-Base (86M) for enhanced attention computation [33,34]. Additionally, we employed EfficientNet B3 (12M), MobileViT (5.6M), and Xception (22M) for efficient scaling, MobileNet integration, and depthwise separable convolutions [35].

**Graph Construction:** The graph construction process begins by defining the **nodes** of the graph. Each node corresponds to a feature vector **z**, representing the image's key characteristics after feature extraction. The set of nodes is denoted as $\mathcal{V} = \{\mathbf{z}\}$, where each node represents an individual image's feature vector. Next, the **edges** between nodes are defined based on the distances between their corresponding feature vectors. These distances are a combination of **spatial distance**, $d_{\mathrm{sp}}(i,j)$, and semantic distance, $d_{\mathrm{se}}(i,j)$. The combined distance between two nodes $i$ and $j$ is computed as:

$$d_{\mathrm{comb}}(i,j) = \beta \cdot d_{\mathrm{sp}}(i,j) + (1 - \beta) \cdot d_{\mathrm{se}}(i,j), \beta \in [0,1] \tag{2}$$

Here, $\beta$ is a hyperparameter that controls the weighting between spatial and semantic distances. The resulting graph is represented as $G = (\mathcal{V}, \mathcal{E})$, where $\mathcal{V}$ are the nodes and $\mathcal{E}$ are the edges, with each edge carrying the combined distance weight between nodes. The graph is constructed following algorithm 1.

### Algorithm 1 Graph construction and GCN refinement.

1: **Input:** FV $\mathbf{z} \in \mathbb{R}^d$, graph parameters $\beta$, number of graph convolution layers $L$, graph neighborhood $\mathcal{N}(i)$, GCN weights $\mathbf{W}_l$, biases $\mathbf{b}_l$
2: **Step 1: Graph Construction**
3: Construct graph $G = (\mathcal{V}, \mathcal{E})$
4: Nodes $\mathcal{V} = \{\mathbf{z}\}$
5: **Initialize node embeddings:** $\mathbf{h}_i^{(0)} = \mathbf{z}_i$
6: Compute combined distance $d_{\mathrm{comb}}(i,j)$ using:

$$d_{\mathrm{comb}}(i,j) = \beta \cdot d_{\mathrm{sp}}(i,j) + (1 - \beta) \cdot d_{\mathrm{se}}(i,j), \quad \beta \in [0,1]$$

 where $d_{\mathrm{sp}}(i,j)$ is the spatial distance and $d_{\mathrm{se}}(i,j)$ is the semantic distance.
7: **Step 2: GCN Refinement**
8: **for** layer $l = 1$ to $L$ **do**
9: **for** node $i$ **do**
10: $\mathbf{h}_i^{(l+1)} \leftarrow \sigma\left(\sum_{j \in \mathcal{N}(i)} \frac{1}{\sqrt{\deg(i)\deg(j)}} \mathbf{W}_l \mathbf{h}_j^{(l)} + \mathbf{b}_l\right)$
11: **end for**
12: **end for**
13: **Step 3: Output**
14: Return the final node embeddings $\mathbf{h}^{(L)}$

**Graph Convolutional Network (GCN):** GCN operates by refining node embeddings through multiple layers, where each layer aggregates information from neighboring nodes. At each layer, the embedding of a node $\mathbf{h}_i^{(l+1)}$ is updated by performing a weighted sum of the embeddings of its neighbors $\mathbf{h}_j^{(l)}$, normalized by the degree of the nodes. This update rule is given by:

$$\mathbf{h}_i^{(l+1)} = \sigma\left(\sum_{j \in \mathcal{N}(i)} \frac{1}{\sqrt{\deg(i)\deg(j)}} \mathbf{W}_l \mathbf{h}_j^{(l)} + \mathbf{b}_l\right) \tag{3}$$

where $\mathbf{W}_l$ and $\mathbf{b}_l$ are learnable weights and biases for layer $l$, $\sigma$ is a nonlinear activation function (typically ReLU), and $\deg(i)$ is the degree of node $i$. The normalization term ensures that nodes with higher degrees do not dominate the aggregation process. The GCN operates over $L$ layers, where each successive layer aggregates information from nodes that are increasingly further away in the graph, allowing each node's embedding to incorporate more global information. After $L$ layers, the node embeddings $\mathbf{h}_i^{(L)}$ capture both local and global structural information of the graph. These refined embeddings are then used for multiclass DR prediction.

### 3.5 Tuning pipeline

FV is passed to the constructed GCN block. The GCN is applied to refine the final node embeddings $\mathbf{h}_i^{(l+1)}$. After refining the embeddings, a classification head is used to obtain the predicted label $\hat{\mathbf{y}}$ Alongside classification, the predicted quality $\hat{q}$ is computed, and the QA loss $\mathcal{L}_q$ is calculated. The **total loss** is a combination of the classification loss $\mathcal{L}_{cls}$ and quality assessment loss $\mathcal{L}_q$, weighted by the hyperparameters $\lambda : \mathcal{L}_{total} = \mathcal{L}_{cls} + \lambda\mathcal{L}_q$ Finally, the model parameters $\Theta$ are updated by minimizing the total loss: $\Theta \leftarrow \Theta - \eta\nabla_\Theta\mathcal{L}$ where $\eta$ is the learning rate. This process is repeated iteratively for each training batch to optimize the model. The training pipeline is given in algorithm 2.

**Algorithm 2 Training pipeline.**

```
 1: Input: Training dataset {(xᵢ,yᵢ)}ᴺᵢ₌₁, learning rate η, epochs E, regularization λ, stochastic passes T
 2: Initialize model parameters Θ
 3: for epoch = 1 to E do
 4:   for batch = 1 to num_batches do
 5:     Sample mini-batch {xᵢ,yᵢ}ᴮᵢ₌₁
 6:     for each image xᵢ do
 7:       Extract features: zᵢ = f(xᵢ)
 8:       Call Graph_GCN(zᵢ) to get hᵢ⁽ᴸ⁾
 9:       Classification:
10:       Predict: ŷᵢ = softmax(W_clf hᵢ⁽ᴸ⁾)
11:       Perform uncertainty estimation: ȳᵢ,σᵢ²
12:       Compute loss: L = L_cls + λL_q
13:     end for
14:     Backpropagate: ∇_ΘL
15:     Update parameters: Θ ← Θ - η∇_ΘL
16:   end for
17: end for
18: Output: Optimized Θ
```

## 4 Experiment

In this section, we provide a comprehensive overview of our experimental setup, detailing the hardware and software configurations, dataset preprocessing, and training procedures. We then present the results obtained from our experiments, followed by an in-depth discussion analyzing the performance of different approaches.

## 4.1 Setup

In this section, we outline the setup criteria for our experiments, including dataset preparation, model configurations, and evaluation strategies. We describe the preprocessing steps applied to the datasets, the architectures and hyperparameters used for training the models, and the metrics employed to assess their performance.

**Dataset:** To prepare the dataset for training, we balanced the samples across the five DR classes by oversampling and augmenting minority class images. First, let $N_{max}$ be the number of samples in the majority class. We oversampled the minority classes by duplicating and augmenting their images to match $N_{max}$. Augmentation was done using the Albumentations pipeline with transformations such as random rotations, flips, blur operations, and brightness/contrast adjustments. All of the images were resized to $224 \times 224$. The original and augmented samples were combined to maintain class balance. Finally, the dataset was split into training (70%), validation (15%), and test (15%) sets using stratified splitting to preserve class distributions. This balanced dataset was then used for model training and evaluation.

**Implementation Details:** In our implementation, we employ a GCN technique, which takes a graph as input with $k = 4$ nearest neighbors and $radius = 0.1$ for creating the graph edges. During evaluation, the model performs $T = 10$ Monte Carlo dropout passes with dropout rates of $p = 0.3$ in the classifier and $p = 0.2$ in the GCN. Random seeds were fixed to 42 for PyTorch, CUDA, and NumPy to ensure exact reproducibility. We trained this model with the AdamW optimizer using an initial learning rate of $5e^{-5}$ and weight decay of 0.01. The cross-entropy loss is employed as $\mathcal{L}_{cls}$ with a hyperparameter $\lambda = 0.1$. It is worth mentioning that various hyperparameters of the applied models are automatically tuned employing the Optuna framework. We have used a ReduceLROnPlateau scheduler with a patience of 7 and a minimum learning rate of $1e^{-7}$. Early stopping is employed with a patience of 15 epochs, a total of 50 epochs, to avoid overfitting. Table 3 depicts the detailed hyperparameters used in this experiment. It categorizes parameters into training configuration, graph construction, Uncertainty Estimation, GCN configuration, and loss function. Key details include batch size, learning rate, optimizer, graph parameters, and loss weights, ensuring an optimized model training process with efficient learning and generalization. Table 4 represents the training parameters and time analysis for both datasets (APTOS2019 and Messidor-2) during experiments. All experiments were performed on an NVIDIA GeForce RTX 4070 GPU with 12GB of VRAM, using the PyTorch framework. We used 32 images as the batch size for balancing memory constraints and training efficiency. The implementation codes can be found at: https://github.com/mfar201/diabetic_retinopathy_classification_gcn.

**Table 3**. Hyperparameter values used in our experiments.

| Category | Hyperparameter | Value | Description |
|---|---|---|---|
| Training Configuration | Number of Epochs | 50 | Total number of training epochs |
|  | Batch Size | 32 | Samples per batch during training |
|  | Learning Rate | 5e-5 | Initial learning rate |
|  | Optimizer | AdamW | Optimization algorithm |
|  | Weight Decay | 0.01 | L2 regularization parameter) |
|  | Learning Rate Scheduler | ReduceLROnPlateau | Adjusts learning rate based on validation loss |
|  | Scheduler Patience | 7 | Epochs with no improvement before reducing LR |
|  | Early Stopping Patience | 15 | Epochs with no improvement before stopping training |
| Graph Construction | Number of Neighbors ($k$) | 4 | Nearest neighbors in graph construction |
|  | Radius | 0.1 | Threshold for edge creation |
|  | Feature Weight | 0.5 | Balance between spatial and semantic features |
| Uncertainty Estimation | MC Dropout Samples | 10 | Number of forward passes for Monte Carlo dropout |
|  | Inference Dropout Rate | 0.3 | Dropout rate used during uncertainty estimation |
| GCN Configuration | Input Dimension | 1024 | Dimension of input features |
|  | Hidden Dimensions | [512, 256] | Sizes of hidden layers |
|  | Output Dimension | 1024 | Dimension of GCN output features |
|  | Dropout Rate | 0.2 | Dropout applied to prevent overfitting |
| Loss Function | Classification Loss Weight | 1.0 | Weight for classification loss |
|  | Quality Assessment Loss Weight | 0.1 | Weight for quality assessment loss |

PLOS Computational Biology

**Table 4. Model-specific training parameters.** The number of parameters refers to trainable parameters in the backbone. Single Image Inference Time (SIIT) is measured in milliseconds (ms), and Per Epoch (TTPE) is measured in seconds (s). These properties are independent of datasets.

| Architecture | Backbone | Model Size (MB) | # of Parameters | SIIT (ms) | IIPE (s) |
|---|---|---|---|---|---|
| **CNN** | ResNet50 | 295.27 | 25,755,462 | 16.59 | 88.95 |
| | ResNet101 | 513.07 | 44,747,590 | 19.23 | 95.61 |
| | ResNet152 | 692.52 | 60,391,238 | 22.82 | 1138.60 |
| | Xception | 458.76 | 40,017,990 | 15.26 | 1458.57 |
| | InceptionV3 | 275.69 | 24,032,998 | 18.88 | 93.30 |
| | InceptionResNetV2 | 642.58 | 56,025,510 | 36.20 | 100.40 |
| | EfficientNetB3 | 142.99 | 12,415,278 | 17.82 | 99.00 |
| | DenseNet121 | 94.20 | 8,144,518 | 20.92 | 114.47 |
| | DenseNet161 | 332.29 | 28,884,550 | 26.13 | 103.75 |
| | DenseNet169 | 165.58 | 14,335,622 | 23.34 | 97.11 |
| | DenseNet201 | 233.23 | 20,208,262 | 28.93 | 105.91 |
| **Transformer** | ViT-Base | 992.76 | 86,725,126 | 15.65 | 104.87 |
| | Swin-Base | 1006.84 | 87,933,886 | 24.36 | 101.43 |
| | DeiT-Base | 992.77 | 86,726,662 | 13.49 | 91.84 |
| | MobileViT | 28.60 | 2,463,030 | 19.22 | 98.91 |

**Evaluation metrics:** We measure the performance of our model using five metrics: accuracy (Acc) for overall performance; macro-averaged F1-score (F1) for per-class effectiveness; Cohen's Kappa for chance-adjusted agreement; Area Under the Receiver Operating Characteristic Curve (AUROC) for classification capability; and Area Under the Precision-Recall Curve (AUPR) for handling class imbalance. For justifications, we analyzed confusion matrices and precision-recall curves.

**Comparative Analysis of Preprocessing Techniques:** A central claim of our work is that our proposed GCN-enhanced framework performs robustly without requiring extensive image preprocessing. To support this claim, we designed experiments to compare our primary pipeline against two widely used preprocessing methods for DR classification. These methods were applied to the entire dataset before the training process and were evaluated separately from our main model.

**(a) CLAHE:** Contrast Limited Adaptive Histogram Equalization (CLAHE) is a contrast enhancement algorithm used to improve image contrast while preventing over-amplification. It enhances local contrast and is particularly effective in highlighting features in homogeneous regions [36].

**(b) Ben-Graham Method:** Ben Graham, a researcher in the deep learning domain, devised a preprocessing technique often used in medical image analysis tasks to improve images with varying lighting conditions, noise, or imbalance in contrast. This algorithm is used to enhance the features of the retinal fundus images and make the dataset more uniform by handling the variations in the brightness of the images [37].

The results of this comparative analysis, presented in Figs 3, 4, and 5 and Table 5, evaluate the performance of our model under three conditions: (1) our proposed pipeline with no advanced preprocessing, (2) with CLAHE preprocessing, and (3) with Ben Graham preprocessing.

## 5 Results

We have compared our model with existing state-of-the-art (SOTA) approaches, highlighting improvements in accuracy, robustness, and reliability. Additionally, we assess the impact of our novel loss formulation, including classification loss and QA loss, on model performance. To ensure a comprehensive evaluation, we provide detailed quantitative results in Table 5, Table 6, and visual interpretations using GradCAM in Fig 5.

We evaluated our proposed framework under three distinct preprocessing conditions to assess the impact of these techniques on performance. The conditions were: our primary pipeline using only basic resizing and normalization, a pipeline incorporating CLAHE, and a pipeline using the Ben-Graham method. The following results compare our primary approach

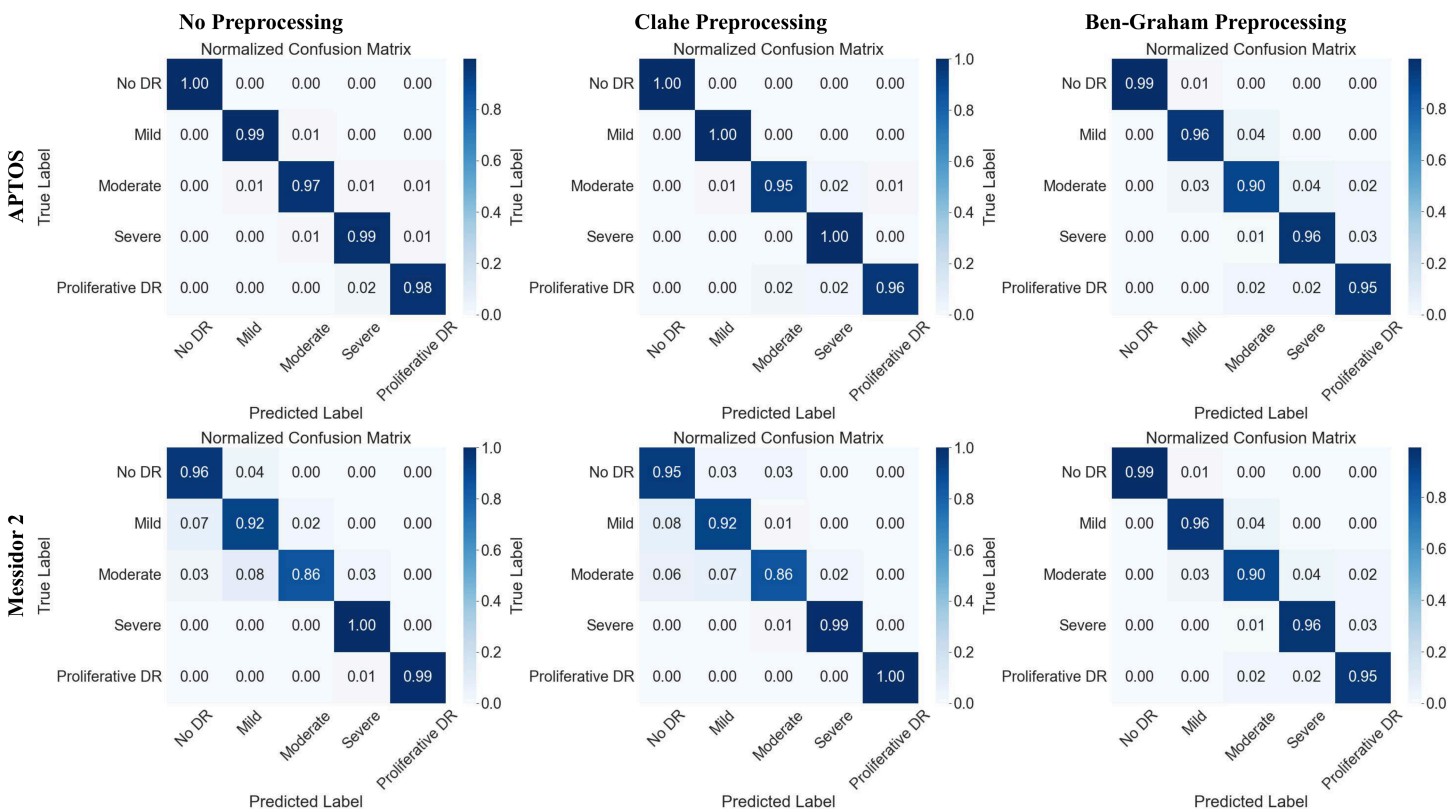

**Fig 3**. **Comparison of normalized confusion matrices for multiclass DR classification on the APTOS and Messidor-2 datasets using different preprocessing methods: Our pipeline with No Preprocessing (left), CLAHE preprocessing (middle), and Ben-Graham preprocessing (right). MobileViT** model on *APTOS2019* dataset shows excellent performance, with minimal misclassification and high precision-recall (AP=1.00). **DenseNet169** on the *Messidor-2* dataset achieves high accuracy.

with existing state-of-the-art (SOTA) methods and analyze its performance relative to the preprocessing-intensive variants.

**Performance Comparison:** After training the models on a balanced dataset, we evaluated their performance on the actual test sets of APTOS2019 and Messidor-2, where the data distribution is inherently imbalanced. This evaluation ensures the model's generalizability to real-world scenarios. We have compared our models with all of the SOTA pipelines. Table 5 presents a comparative analysis of various deep-learning models for DR grading on the APTOS2019 and Messidor-2 datasets, showcasing results from existing studies alongside the authors' proposed models. Among CNN-based architectures, DenseNet169 achieves the highest accuracy (94.51%) and F1-score (94.49%) on Messidor-2, while MobileViT outperforms other models on APTOS2019, achieving the highest accuracy (98.45%) and AUROC (0.9994). The proposed models, particularly MobileViT and DenseNet variants, consistently surpass prior CNN and transformer architectures, demonstrating improved classification performance across both datasets. Table 6 presents the external validation performance on the EyePACS dataset using two top-performing backbones—DenseNet-169 (pretrained on Messidor-2) and MobileViT (pretrained on APTOS2019). After fine-tuning on EyePACS, both models demonstrated consistently high performance across all evaluation metrics: Accuracy, F1-score, AUROC, AUPR, and Cohen's $\kappa$. The applied DenseNet-169 technique achieved the best performance, with 97.38% accuracy and 99.83% AUROC. The demonstrated results suggest that the proposed framework is robust and transferable across datasets, addressing the common concern of limited dataset dependency in previous studies.

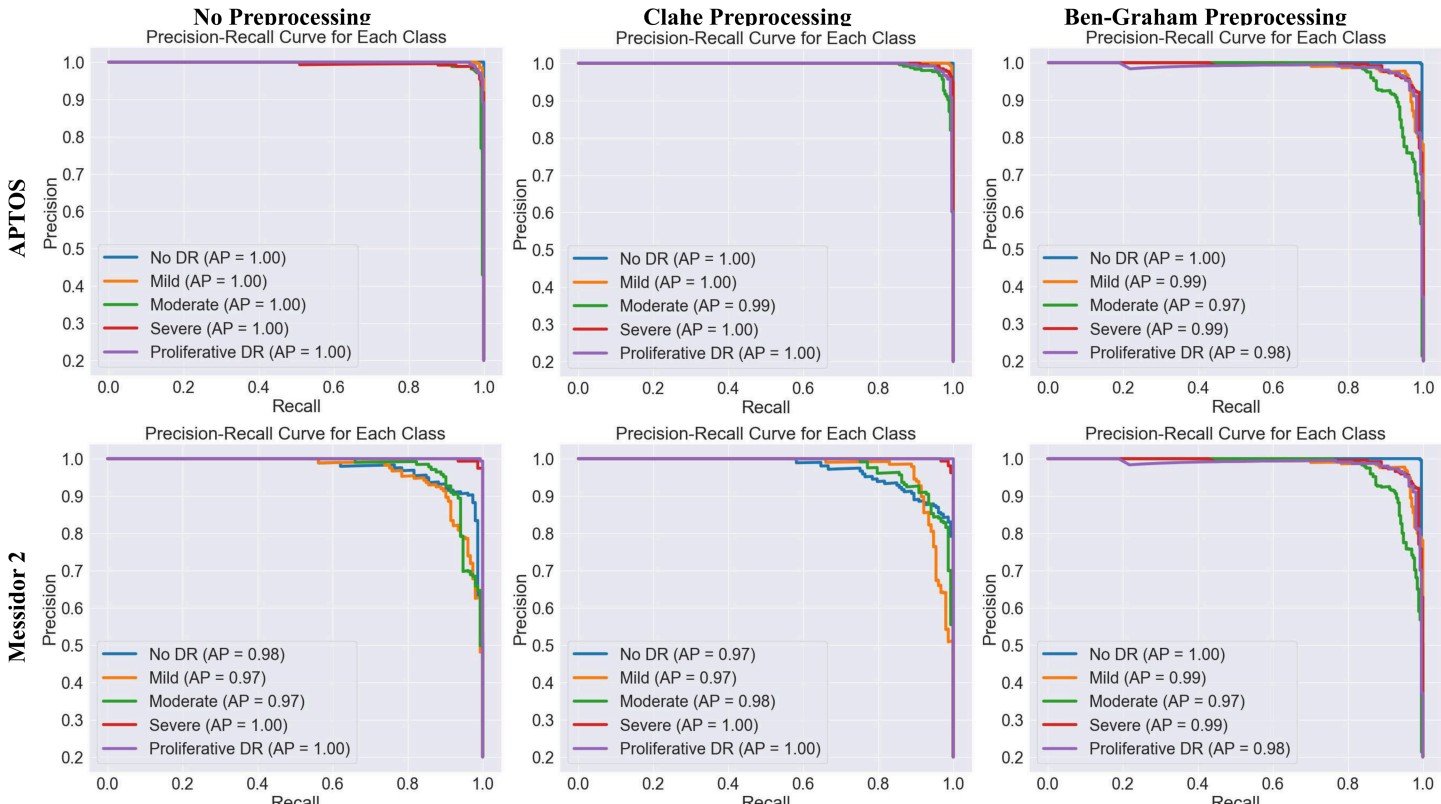

**Fig 4**. Precision-recall curves for multiclass DR classification on the APTOS and Messidor-2 datasets using three different preprocessing methods: our pipeline with no preprocessing (left), CLAHE Preprocessing (middle), and Ben-Graham preprocessing (right). MobileViT model on *APTOS2019* dataset shows excellent performance, with minimal misclassification and high precision-recall (AP=1.00). **DenseNet169** on the *Messidor-2* dataset achieves high accuracy.

**Explainable AI Visualization:** To ensure the model's predictions are not only accurate but also interpretable, Grad-CAM is employed to visualize the regions influencing its classification decisions, as shown in Fig 5. A detailed analysis of these heatmaps reveals that the model has learned to identify clinically relevant pathologies and that its focus correctly shifts in alignment with the increasing severity of Diabetic Retinopathy.

- **No DR (Class 0):** For fundus images of healthy retinas, the Grad-CAM activations are diffuse and lack a specific focus. This indicates the model is confirming the absence of key pathological markers, which is the desired behavior for a negative diagnosis.
- **Mild NPDR (Class 1):** In this early stage, the model's attention is drawn to small, punctate areas of high activation. These highlighted spots indicate the emergence of microaneurysms, which are the earliest signs of DR.
- **Moderate NPDR (Class 2):** As the disease progresses to the moderate stage, the activated regions on the heatmaps become larger and more pronounced. This shift in focus aligns with the clinical presentation of dot and blot hemorrhages and hard exudates, which are more significant vascular lesions than microaneurysms.
- **Severe NPDR & Proliferative DR (PDR) (Class 3 and 4):** In the most advanced stages, the Grad-CAM visualizations show large, intense areas of activation. These regions often correspond to significant retinal hemorrhages and, crucially, areas of neovascularization (the growth of new, abnormal blood vessels). The model's focus on these features,

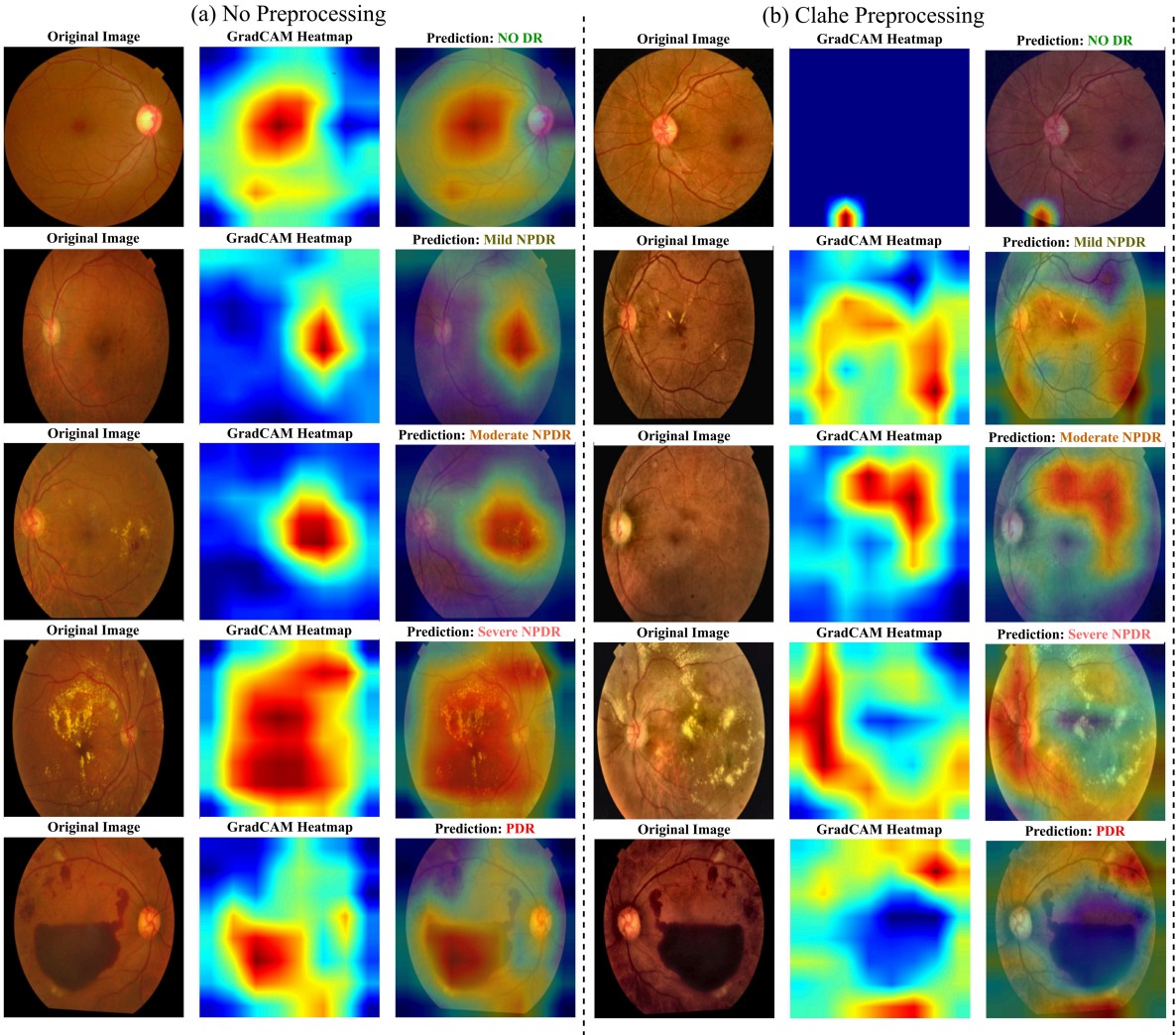

**Fig 5**. **Grad-CAM heatmaps were generated for retinal fundus images in the DR classification task.** Each row presents the original image, its corresponding Grad-CAM heatmap, and the model's prediction. The red regions in the heatmaps indicate areas that strongly influence the classification. (a) represents our approach without any sophisticated preprocessing techniques, which performs significantly better than the other two: (b) with CLAHE preprocessing.

which are the indications of severe and proliferative DR, demonstrates its ability to recognize the most critical, vision-threatening signs of the disease.

This stage-by-stage analysis confirms that this framework bases its decisions on recognized clinical indicators of DR. The progressive shift in the model's attention from minor to major pathologies provides strong evidence of its clinical relevance and enhances trust in its utility as a reliable diagnostic tool.

**Analysis:** The GCN-based framework with MobileViT and DenseNet169 consistently improves DR classification accuracy, showcasing its effectiveness in ophthalmological studies. Fig 3 illustrates strong classification performance with minimal misclassification in both datasets. The confusion matrix reveals accurate predictions for the No DR and Moderate DR classes, while the precision-recall curves show an average precision (AP) of 1.00 across all categories for the APTOS

**Table 5**. Comparison with SOTA: After training the models using three different approaches—(1) applying CLAHE (♣), (2) applying Ben Graham's preprocessing technique (♦), and (3) our proposed (♡) pipeline without sophisticated preprocessing—we evaluated them on the actual test sets of the APTOS2019 and Messidor-2 datasets. Our approach outperformed all existing benchmarks for DR classification on both datasets. Whether using CNN (■) or Transformer architectures (▲), our method consistently achieved superior performance compared to all previous DR classification methods.

| Arch. | Meth. | Backbone | APTOS2019 | | | | | Messidor-2 | | | | |
|---|---|---|---|---|---|---|---|---|---|---|---|---|
| | | | Acc(%) | F1(%) | AUROC(%) | AUPR(%) | Kappa(%) | Acc(%) | F1(%) | AUROC(%) | AUPR(%) | Kappa(%) |
| ■ | [6] | ResNet50 | 84.80 | 84.30 | - | - | 90.90 | 67.10 | 65.50 | - | - | 66.30 |
| | [6] | DenseNet121 | 85.50 | 84.90 | - | - | 90.60 | 68.00 | 66.00 | - | - | 67.30 |
| | [17] | DenseNet121 | 97.30 | - | - | - | - | - | - | - | - | - |
| | [18] | RSG-Net | - | - | - | - | - | 99.36 | 99.40 | 99.98 | 99.40 | - |
| | [21] | ResNet50 | 85.65 | - | 89.00 | - | - | - | - | - | - | - |
| | [38] | DenseNet201 | 91.62 | 91.52 | - | - | - | 85.79 | 85.08 | - | - | - |
| | [38] | MobileNetV2 | 93.09 | 93.53 | - | - | - | 83.81 | 85.23 | - | - | - |
| | [39] | MobileNetV2 | 92.00 | - | - | - | - | - | - | - | - | - |
| | [40] | DenseNet121 | 97.68 | 97.00 | 96.20 | - | 98.50 | - | - | - | - | - |
| | [41] | ResNet152 | 85.94 | 71.29 | 82.96 | - | - | - | - | - | - | - |
| | [18,42] | RSG-Net | - | - | - | - | - | 99.36 | - | - | - | - |
| | [43] | Xception | 84.36 | 70.49 | 93.82 | - | - | 74.21 | 55.18 | 87.26 | - | - |
| ▲ | [44] | Swin-Base | 88.70 | 88.70 | - | - | - | 83.12 | 83.12 | - | - | - |
| | [45] | Swin-Base | 86.40 | - | - | - | - | - | - | - | - | - |
| ■ | ♣ | ResNet50 | 94.61 | 94.58 | 99.52 | 98.34 | 93.27 | 75.69 | 75.89 | 96.54 | 87.86 | 69.61 |
| | | ResNet101 | 80.52 | 79.81 | 98.78 | 96.02 | 75.65 | 37.91 | 27.37 | 81.48 | 60.93 | 22.39 |
| | | ResNet152 | 92.10 | 92.16 | 99.23 | 97.39 | 90.13 | 81.57 | 81.45 | 0.9780 | 93.76 | 76.96 |
| | | Xception | 95.42 | 95.42 | 99.68 | 98.95 | 94.28 | 94.12 | 94.14 | 99.33 | 97.94 | 92.65 |
| | | InceptionV3 | 96.53 | 96.53 | 99.78 | 99.25 | 95.66 | 92.29 | 92.22 | 99.22 | 97.55 | 90.36 |
| | | InceptionResNetV2 | 96.38 | 96.37 | 99.63 | 98.75 | 95.48 | 93.20 | 93.17 | 99.47 | 98.17 | 91.50 |
| | | EfficientNetB3 | 96.97 | 96.98 | 99.84 | 99.45 | 96.22 | 92.94 | 92.98 | 99.41 | 98.11 | 91.18 |
| | | DenseNet121 | 90.11 | 90.02 | 99.70 | 99.02 | 87.64 | 85.62 | 85.43 | 98.64 | 95.85 | 82.03 |
| | | DenseNet161 | 95.50 | 95.48 | 99.71 | 99.02 | 94.37 | 93.46 | 93.46 | 99.40 | 98.02 | 91.83 |
| | | DenseNet169 | 96.68 | 96.68 | 99.75 | 99.20 | 95.85 | 93.46 | 93.46 | 99.34 | 97.84 | 91.83 |
| | | DenseNet201 | 95.65 | 95.63 | 99.78 | 99.23 | 94.56 | 94.25 | 94.23 | 99.53 | 98.44 | 92.81 |
| ▲ | ♣ | ViT-Base | 95.57 | 95.56 | 99.51 | 98.51 | 94.46 | 89.41 | 89.41 | 98.54 | 95.71 | 86.76 |
| | | Swin-Base | 96.90 | 96.89 | 99.81 | 99.36 | 96.13 | 91.24 | 91.28 | 98.96 | 96.69 | 89.05 |
| | | DeiT-Base | 96.01 | 96.00 | 99.69 | 99.01 | 95.02 | 91.90 | 91.87 | 99.08 | 97.02 | 89.87 |
| | | MobileViT | 98.01 | 98.00 | 99.94 | 99.79 | 97.51 | 91.37 | 91.43 | 99.03 | 96.67 | 89.22 |
| ■ | ♦ | ResNet50 | 81.99 | 81.99 | 96.59 | 89.36 | 77.49 | 68.37 | 67.01 | 93.62 | 77.77 | 60.46 |
| | | ResNet101 | 70.63 | 69.56 | 93.71 | 83.85 | 63.28 | 20.00 | 6.670 | 62.60 | 30.70 | 00.00 |
| | | ResNet152 | 78.67 | 77.63 | 94.88 | 88.10 | 73.34 | 54.12 | 49.96 | 86.21 | 63.66 | 42.65 |
| | | Xception | 93.21 | 93.15 | 99.16 | 97.50 | 91.51 | 93.46 | 93.43 | 99.00 | 96.91 | 91.83 |
| | | InceptionV3 | 94.39 | 94.34 | 99.61 | 98.72 | 92.99 | 93.59 | 93.59 | 99.48 | 98.27 | 91.99 |
| | | InceptionResNetV2 | 90.04 | 90.10 | 99.14 | 97.33 | 87.55 | 92.03 | 91.96 | 99.10 | 97.05 | 90.03 |
| | | EfficientNetB3 | 93.21 | 93.16 | 99.48 | 98.29 | 91.51 | 91.50 | 91.43 | 98.78 | 96.22 | 89.38 |
| | | DenseNet121 | 89.08 | 89.15 | 98.62 | 95.83 | 86.35 | 85.62 | 85.64 | 98.12 | 94.73 | 82.03 |
| | | DenseNet161 | 95.13 | 95.13 | 99.63 | 98.80 | 93.91 | 94.64 | 94.62 | 99.50 | 98.34 | 93.30 |
| | | DenseNet201 | 94.17 | 94.16 | 99.37 | 97.97 | 92.71 | 93.73 | 93.70 | 99.57 | 98.49 | 92.16 |
| | | DenseNet169 | 94.83 | 94.78 | 99.55 | 98.56 | 93.54 | 95.03 | 95.01 | 99.50 | 98.32 | 93.79 |
| ▲ | ♦ | ViT-Base | 94.39 | 94.37 | 99.34 | 98.08 | 92.99 | 90.33 | 90.24 | 98.42 | 95.08 | 87.91 |
| | | Swin-Base | 95.42 | 95.42 | 99.58 | 98.73 | 94.28 | 90.59 | 90.62 | 98.73 | 96.00 | 88.24 |
| | | DeiT-Base | 94.98 | 94.95 | 99.51 | 98.48 | 93.73 | 90.72 | 90.70 | 98.69 | 95.90 | 88.40 |
| | | MobileViT | 93.95 | 93.95 | 99.22 | 97.31 | 92.44 | 92.55 | 92.57 | 99.31 | 97.68 | 90.69 |
| ■ | ♡ | ResNet50 | 93.95 | 94.02 | 99.70 | 98.98 | 92.44 | 70.46 | 67.28 | 92.55 | 78.90 | 63.07 |
| | | ResNet101 | 84.65 | 84.04 | 97.79 | 93.54 | 80.81 | 40.00 | 22.86 | 79.60 | 54.07 | 25.00 |
| | | ResNet152 | 92.18 | 92.13 | 99.25 | 97.67 | 90.22 | 79.87 | 80.07 | 96.53 | 90.61 | 74.84 |
| | | Xception | 95.28 | 95.26 | 99.59 | 98.67 | 94.10 | 93.07 | 93.05 | 99.14 | 97.34 | 91.34 |
| | | InceptionV3 | 96.97 | 96.97 | 99.82 | 99.38 | 96.22 | 94.12 | 94.08 | 99.49 | 98.35 | 92.65 |
| | | InceptionResNetV2 | 94.98 | 94.95 | 99.41 | 97.97 | 93.73 | 94.38 | 94.37 | 99.37 | 97.95 | 92.97 |
| | | EfficientNetB3 | 96.83 | 96.82 | 99.83 | 99.40 | 96.03 | 94.12 | 94.12 | 99.52 | 98.42 | 92.65 |
| | | DenseNet121 | 93.87 | 93.80 | 99.44 | 98.12 | 92.34 | 87.58 | 87.59 | 98.65 | 96.06 | 84.48 |
| | | DenseNet161 | 96.24 | 96.23 | 99.70 | 99.02 | 95.30 | 94.38 | 94.39 | 99.52 | 98.41 | 92.97 |
| | | DenseNet201 | 96.61 | 96.60 | 99.84 | 99.43 | 95.76 | 92.94 | 92.92 | 99.35 | 97.90 | 91.18 |
| | | DenseNet169 | 96.75 | 96.74 | 99.81 | 99.37 | 95.94 | **94.90** | **94.87** | 99.50 | **98.42** | **93.63** |
| ▲ | ♡ | ViT-Base | 95.57 | 95.55 | 99.62 | 98.79 | 94.46 | 88.63 | 88.66 | 98.06 | 94.17 | 85.78 |
| | | Swin-Base | 96.90 | 96.89 | 99.83 | 99.44 | 96.13 | 91.24 | 91.23 | 99.04 | 96.85 | 89.05 |
| | | DeiT-Base | 95.72 | 95.71 | 99.71 | 99.07 | 94.65 | 90.20 | 90.17 | 98.91 | 96.52 | 87.75 |
| | | MobileViT | **98.45** | **98.45** | **99.94** | **99.81** | **98.06** | 92.03 | 92.02 | 99.25 | 97.45 | 90.03 |

**Table 6**. External validation on EyePACS: Performance of the two best backbones from our pipeline—DenseNet-169 (initially optimized on Messidor-2) and MobileViT (initially optimized on APTOS2019)—after a brief fine-tuning stage on the EyePACS dataset.

| Initially Trained on | Backbone | EyePACS | | | | |
|---|---|---|---|---|---|---|
| | | Acc(%) | F1(%) | AUROC(%) | AUPR(%) | Kappa(%) |
| Messidor-2 | DenseNet169 | 97.38 | 97.37 | 99.83 | 99.39 | 96.72 |
| APTOS2019 | MobileViT | 96.02 | 96.02 | 99.60 | 98.69 | 95.03 |

dataset. Similarly, for Messidor-2, the best model achieves high accuracy despite some misclassification, as shown in Fig 4.

## 5.1 Ablation study

Out of several experiments, our primary focus was on evaluating the impact of the imbalanced dataset and the choice of optimizer. Fig 6. presents a performance comparison for the APTOS2019 and Messidor-2 datasets using the MobileViT and DenseNet169 backbones, respectively.

**Impact of Imbalanced Datasets:** Initially, we experimented with the original imbalanced dataset (OgD). As shown in Fig 6(a), the models demonstrated the lowest performance with the OgD. To address this issue, we applied two class balancing strategies: "Compute Class Weight" (OgD WC) and "Weighted Random Sampler" (OgD RS). Despite these balancing techniques, Fig 6(a) demonstrates that our approach outperformed all other strategies across all metrics.

**Impact of Optimizer:** We also observed a decline in model performance when switching from AdamW to SGD as the optimizer. This scenario is illustrated in Fig 6(b).

## 6 Discussion

This study demonstrates the efficacy of DL models in diagnosing and grading DR. Classification accuracy and AUC scores are improved when CNNs are used for spatial feature extraction and ViTs are used for global context. The model outperforms conventional CNN techniques by exhibiting high sensitivity and specificity through ROC curves and confusion matrices. Explainable AI techniques, such as Grad-CAM, improve transparency and trust in clinical applications. However, challenges remain, including dependence on high-quality labeled data and computational complexity, limiting real-time

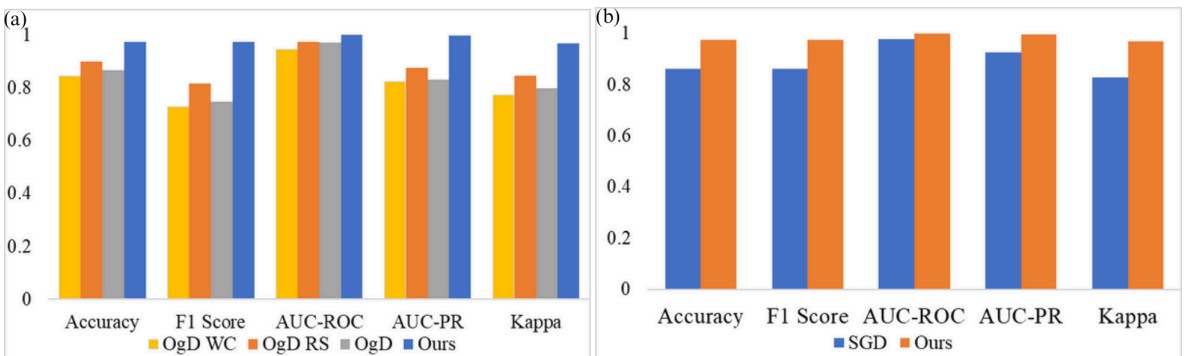

**Fig 6**. (a) Training strategies were applied to OgD WC, OgD RS, and OgD, with the proposed method evaluated on APTOS. Our approach (black bar) consistently achieves the highest accuracy, F1-score, AUROC, AUPR, and Kappa, demonstrating superior classification performance and agreement. (b) In comparison, our proposed approach outperforms SGD across all metrics, highlighting its better generalization and robustness.

deployment. Future research should optimize efficiency, incorporate multimodal data, e.g., optical coherence tomography (OCT), and enhance interpretability through saliency maps and attention mechanisms for broader clinical applicability.

**Limitations:** Our training is conducted on an augmented dataset, which is justified for experimental purposes. However, incorporating real-world retinal images would enable the models to learn actual DR patterns, leading to more accurate and reliable predictions. Additionally, emerging vision-language models (VLMs) and ensemble-based approaches remain unexplored, which could further enhance classification performance.

## 7 Conclusions

This study emphasizes ethical integrity in developing and evaluating a DR classification framework using GCNs. A novel approach has been developed for DR classification using pre-trained models for feature extraction, followed by Graph Convolutional Networks (GCNs) to refine embeddings. The extracted feature vectors are structured as a graph, where GCN enhances embeddings before classification, and a quality assessment module predicts a confidence score using a fully connected layer trained with binary cross-entropy loss. Uncertainty estimation is performed by calculating the variance across multiple stochastic passes, providing a measure of prediction reliability. The proposed method is evaluated on the APTOS2019 and Messidor-2 datasets, demonstrating superior performance compared to state-of-the-art methods. Grad-CAM heat maps were employed to improve interpretability and facilitate clinical validation. Furthermore, including the large-scale EyePACS dataset in external validation demonstrates the framework's ability to generalize across diverse imaging conditions, demographics, and grading variations, enhancing robustness and reliability for real-world DR screening. This study aligns with ethical guidelines to promote trustworthy artificial intelligence applications in ophthalmology, thereby facilitating impartial and accurate detection of DR.

**Future work:** We aim to explore Vision-Language Models (VLMs) for enhanced interpretability and ensemble learning for improved robustness. Incorporating real-world retinal images will ensure better generalization, while self-supervised learning can reduce reliance on labeled data. Additionally, advancing uncertainty estimation and explainability tools will further enhance the reliability of AI-assisted DR diagnosis.

## Author contributions

**Conceptualization:** Zarin Akter, Riasat Khan.

**Data curation:** Zarin Akter, Jawad Ibn Ahad, Md. Mutasim Farhan.

**Formal analysis:** Zarin Akter, Jawad Ibn Ahad.

**Investigation:** Zarin Akter, Jawad Ibn Ahad, Md. Mutasim Farhan.

**Methodology:** Zarin Akter, Jawad Ibn Ahad, Md. Mutasim Farhan.

**Supervision:** Riasat Khan.

**Validation:** Zarin Akter, Jawad Ibn Ahad.

**Visualization:** Zarin Akter, Jawad Ibn Ahad, Md. Mutasim Farhan.

**Writing – original draft:** Zarin Akter, Jawad Ibn Ahad, Md. Mutasim Farhan.

**Writing – review & editing:** Riasat Khan.

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
