## [Decision Letter · Decision Letter 0]

16 Jul 2025

PCOMPBIOL-D-25-00649

Graph-Enhanced Deep Learning for Diabetic Retinopathy Diagnosis: A Quality-Aware and Uncertainty-Driven Approach

PLOS Computational Biology

Dear Dr. Khan,

Thank you for submitting your manuscript to PLOS Computational Biology. After careful consideration, we feel that it has merit but does not fully meet PLOS Computational Biology's publication criteria as it currently stands. Therefore, we invite you to submit a revised version of the manuscript that addresses the points raised during the review process.

Please submit your revised manuscript within 60 days Sep 15 2025 11:59PM. If you will need more time than this to complete your revisions, please reply to this message or contact the journal office at ploscompbiol@plos.org. Please include the following items when submitting your revised manuscript:

We look forward to receiving your revised manuscript.

Kind regards,

Piero Fariselli

Academic Editor

PLOS Computational Biology

Jennifer Flegg

Section Editor

PLOS Computational Biology

**Journal Requirements:**

2) <carina-action-element class="ng-star-inserted">Please provide an Author Summary. This should appear in your manuscript between the Abstract (if applicable) and the Introduction, and should be 150-200 words long. The aim should be to make your findings accessible to a wide audience that includes both scientists and non-scientists. Sample summaries can be found on our website under Submission Guidelines:</carina-action-element> 

<carina-action-element class="ng-star-inserted">https://journals.plos.org/</carina-action-element><carina-action-element class="ng-star-inserted">ploscompbiol</carina-action-element><carina-action-element class="ng-star-inserted">/s/submission-guidelines#loc-parts-of-a-submission</carina-action-element>  3) <carina-action-element class="ng-star-inserted" style="color: rgba(0, 0, 0, 0.87); font-family: sans-serif; font-size: 12px;">Please upload all main figures as separate Figure files in .tif or .eps format. For more information about how to convert and format your figure files please see our guidelines: </carina-action-element> <carina-action-element class="ng-star-inserted">https://journals.plos.org/</carina-action-element><carina-action-element class="ng-star-inserted">ploscompbiol</carina-action-element><carina-action-element class="ng-star-inserted">/s/figures</carina-action-element> 

4) <carina-action-element class="ng-star-inserted">Some material included in your submission may be copyrighted. According to PLOSu2019s copyright policy, authors who use figures or other material (e.g., graphics, clipart, maps) from another author or copyright holder must demonstrate or obtain permission to publish this material under the Creative Commons Attribution 4.0 International (CC BY 4.0) License used by PLOS journals. Please closely review the details of PLOSu2019s copyright requirements here: PLOS Licenses and Copyright. If you need to request permissions from a copyright holder, you may use PLOS's Copyright Content Permission form.</carina-action-element> 

<carina-action-element class="ng-star-inserted">Please respond directly to this email and provide any known details concerning your material's license terms and permissions required for reuse, even if you have not yet obtained copyright permissions or are unsure of your material's copyright compatibility. Once you have responded and addressed all other outstanding technical requirements, you may resubmit your manuscript within Editorial Manager. </carina-action-element> <carina-action-element class="ng-star-inserted">Potential Copyright Issues:</carina-action-element> <carina-action-element class="ng-star-inserted"></carina-action-element><carina-action-element class="ng-star-inserted">- Figures 1 and 2. Please confirm whether you drew the images / clip-art within the figure panels by hand. If you did not draw the images, please provide (a) a link to the source of the images or icons and their license / terms of use; or (b) written permission from the copyright holder to publish the images or icons under our CC BY 4.0 license. Alternatively, you may replace the images with open source alternatives. See these open source resources you may use to replace images / clip-art: - https://commons.wikimedia.org - https://openclipart.org/.</carina-action-element> 

**Reviewers' comments:**

Reviewer's Responses to Questions

**Comments to the Authors:**

Reviewer #1: • Add the fullform of CNN and DR in the abstract.

• In the abstract add more information about the results.

• Formatting of the introduction is not correct. Starting is of single column and rest is of two column.

• The introduction should provide a stronger motivation for the study by clearly stating the research gap it addresses.

• Format the paper correctly.

• The objectives of the research should be explicitly outlined at the end of the introduction.

• At the end of the introduction add the organization of the paper.

• In fig 1, add the details for the CNN layer.

• Caption of fig 2, is very long. Try to reduce it.

• In table 2, how the hyper-parameter values are selected.

• In table 4, add the year of publication.

• Author can read the following papers to increase the technical strength of the paper:

Ensemble deep learning and EfficientNet for accurate diagnosis of diabetic retinopathy

A Comprehensive Image Processing Framework for Early Diagnosis of Diabetic Retinopathy

Reviewer #2: This work proposes a deep learning approach of diabetic retinopathy diagnosis with uncertainty estimation along with quality assessment. Even though the approach seems promising, at this stage it is unclear how it particularly innovates state-of-the-art approaches. Points to be addressed:

- The authors in "Related works" section critically refer to previous works like Feng et. al. as they test on limited dataset and not in realistic settings, raising this as one of the significant shortcomings. Nonetheless, the authors themselves test their model with only two standard datasets (APTOS2019 and Messidor-2). An external verification would help overcome the limitations highlighted in existing state-of-the-art methods.

- The role of data augmentation procedures (CLAHE, Ben Graham and rotations, flips, brightness adjustments) is not clear in the results. The authors mention as an advantage of their model (in the Introduction) not carrying out "extensive preprocessing," but they in fact make significant use of such preprocessing operations. It is undefined whether they use such procedures for comparison or as central constituents of their pipeline, defying their claim of "minimal preprocessing.

- Explainability: a very brief explainability analysis is provided. The authors should clearly analyze how such regions change in alignment with severity outcome of DR as well as clearly mention their clinically relevant implication. It is stated in far too little detail.

- There is not enough detail in uncertainty estimation description to reproduce. There is a need for clear description of implementation with exact parameters and seed selection settings.

**Have the authors made all data and (if applicable) computational code underlying the findings in their manuscript fully available?**

Reviewer #1: None

Reviewer #2: None

PLOS authors have the option to publish the peer review history of their article (what does this mean?). If published, this will include your full peer review and any attached files.

Reviewer #1: No

Reviewer #2: No

**Figure resubmission:**
---

## [Decision Letter · Decision Letter 1]

23 Sep 2025

PCOMPBIOL-D-25-00649R1

Graph-Enhanced Deep Learning for Diabetic Retinopathy Diagnosis: A Quality-Aware and Uncertainty-Driven Approach

PLOS Computational Biology

Dear Dr. Khan,

Thank you for submitting your manuscript to PLOS Computational Biology. After careful consideration, we feel that it has merit but does not fully meet PLOS Computational Biology's publication criteria as it currently stands. Therefore, we invite you to submit a revised version of the manuscript that addresses the points raised during the review process.

Please submit your revised manuscript within 30 days Nov 23 2025 11:59PM. If you will need more time than this to complete your revisions, please reply to this message or contact the journal office at ploscompbiol@plos.org. Please include the following items when submitting your revised manuscript:

We look forward to receiving your revised manuscript.

Kind regards,

Piero Fariselli

Academic Editor

PLOS Computational Biology

Jennifer Flegg

Section Editor

PLOS Computational Biology

**Journal Requirements:**

1) Please upload Figure 5 as a separate Figure file in .tif or .eps format. For more information about how to convert and format your figure files please see our guidelines: 

**Reviewers' comments:**

Reviewer's Responses to Questions

Reviewer #1: The author addressed all the previous comments but still some issues are pending

• Check my previous comments. Some comments are not addressed correctly.

• The abstract needs to be rewritten to point out significance and impact of the paper.

• In the related work, it is recommended to refer the contribution made by the researchers and the novelty of the research.

• I recommend that the authors add some more current articles to improve the paper's overall quality. The preparation of a comparative analysis of the current publications on this subject should also be included.

• Avoid presenting with lengthy paragraph.

• The references section lacks key foundational or recent works that are crucial to the topic.

• Ensure that all cited works are relevant and up-to-date to reflect the current state of research.

• A more balanced mix of theoretical and empirical studies in the citations would strengthen the paper.

• Finally, a final proof-reading is highly suggested, in order to correct some typos.

Reviewer #2: I find that the authors have responded well to my concerns. They added an external validation set, which strengthens the consistency and generalizability of their method; indeed the method still performs well. They also better explained the preprocess problem making their statements more clear/robust On the explainability section, they expanded the description and linked it to the clinical interpretability. Regarding the uncertainty estimation part, they explained the environment for their implementation, making it possible to reproduce their approach.

I think now the article is clearer, more complete, and methodologically adequate.

I recommend adding a README file to the GitHub repository to enhance the reproducibility of the work. I would include instructions on installing dependencies, preparing the dataset, running the code, and reproducing the main results.

**Have the authors made all data and (if applicable) computational code underlying the findings in their manuscript fully available?**

Reviewer #1: None

Reviewer #2: None

PLOS authors have the option to publish the peer review history of their article (what does this mean?). If published, this will include your full peer review and any attached files.

Reviewer #1: No

Reviewer #2: No

**Figure resubmission:**
---

## [Decision Letter · Decision Letter 2]

13 Nov 2025

Dear Dr. Khan,

We are pleased to inform you that your manuscript 'Graph-Enhanced Deep Learning for Diabetic Retinopathy Diagnosis: A Quality-Aware and Uncertainty-Driven Approach' has been provisionally accepted for publication in PLOS Computational Biology.

Best regards,

Piero Fariselli

Academic Editor

PLOS Computational Biology

Jennifer Flegg

Section Editor

PLOS Computational Biology

Reviewer's Responses to Questions

**Comments to the Authors:**

Reviewer #2: The authors have fully addressed all comments and implemented all requested changes.

**Have the authors made all data and (if applicable) computational code underlying the findings in their manuscript fully available?**

Reviewer #2: Yes

PLOS authors have the option to publish the peer review history of their article (what does this mean?). If published, this will include your full peer review and any attached files.

Reviewer #2: No

---

## [Editor Report · Acceptance letter]

PCOMPBIOL-D-25-00649R2

Graph-Enhanced Deep Learning for Diabetic Retinopathy Diagnosis: A Quality-Aware and Uncertainty-Driven Approach

Dear Dr Khan,

I am pleased to inform you that your manuscript has been formally accepted for publication in PLOS Computational Biology. Your manuscript is now with our production department and you will be notified of the publication date in due course.

With kind regards,

Anita Estes
